# Characterization of nanoparticles combining polyamine detection with photodynamic therapy

Wenting Li[1], Lingyun Wang [1✉], Tianlei Sun[2], Hao Tang [2], Brian Bui[3], Derong Cao[1], Ruibing Wang [2✉] & Wei Chen [3✉]

Polyamine detection and depletion have been extensively investigated for cancer prevention and treatment. However, the therapeutic efficacy is far from satisfactory, mainly due to a polyamine compensation mechanism from the systemic circulation in the tumor environment. Herein, we explore a new solution for improving polyamine detection as well as a possible consumption therapy based on a new photosensitizer that can efficiently consume polyamines via an irreversible chemical reaction. The new photosensitizer is pyrrolopyrrole*aza*-BODIPY pyridinium salt (**PPAB-PyS**) nanoparticles that can react with the overexpressed polyamine in cancer cells and produce two photosensitizers with enhanced phototoxicity on cancer destruction. Meanwhile, **PPAB-PyS** nanoparticles provide a simultaneous ratiometric fluorescence imaging of intracellular polyamine. This combination polyamine consumption with a chemical reaction provides a new modality to enable polyamine detection along with photodynamic therapy as well as a putative depletion of polyamines for cancer treatment and prevention.

[1] Key Laboratory of Functional Molecular Engineering of Guangdong Province, School of Chemistry and Chemical Engineering, South China University of Technology, Guangzhou, China. [2] State Key Laboratory of Quality Research in Chinese Medicine, Institute of Chinese Medical Sciences, University of Macau, Taipa, Macau, SAR, China. [3] Department of Physics, University of Texas at Arlington, Arlington, TX, USA. ✉email: lingyun@scut.edu.cn; rwang@um.edu.mo; weichen@uta.edu

Polyamines (putrescine, spermidine, and spermine) are low molecular weight organic polycations found in mammalian cells in millimolar concentrations. As positively charged molecules, polyamines interact strongly with nucleic acids, proteins, and chromatin[1]. Polyamines are essential for cell proliferation, growth, migration, transcription, RNA stabilization, translational frameshifting, methylation, and acetylation of histones[2]. In the polyamine biosynthesis pathway, putrescine is formed by the action of ornithine decarboxylase (ODC) and is converted into subsequent polyamines by the addition of aminopropyl groups in reactions catalyzed by spermidine synthase and spermine synthase[3]. Polyamine metabolism and requirements are frequently dysregulated in cancer and other hyperproliferative diseases, leading to elevated polyamine levels for excess transformation and tumor progression[4–9].

Since polyamine depletion results in cytostasis, cancer cells are more sensitive to polyamine depletion than normal cells, which provides a feasible modality on cancer treatment called polyamine depletion therapy[10–14]. Until now, several therapeutic agents to downregulate polyamine levels have been developed to fight against tumor cells (Supplementary Table S1). The first one is to specifically inhibit both the biosynthetic and catabolic enzymatic pathways for polyamine metabolism as the therapeutic intervention strategy. For example, inhibition of ODC (the rate-limiting enzyme in polyamine biosynthesis) by use of difluoromethylornithine (DFMO)[15], polyamine blocking therapy through combining DFMO with exogenous polyamine depletion[16], use of DFMO in chemoprevention[17], and polyamine catabolism as a target for chemoprevention[18], are reported by many groups. Although the direct inhibition of polyamine synthetic enzymes has demonstrated some promise, the results have been less satisfactory in the treatment of advanced cancers. The second one is to use polyamine analogs and polyamine conjugates as backbones for prodrug nanoparticles targeting polyamine transport system[19–21]. They can bind to the critical sites normally occupied by polyamines but fail to produce the normal and essential polyamine mediated functions. In addition, they trigger polyamine degradative pathways to reduce the normal polyamine content needed for cell growth. Thirdly, supramolecular traps for catching polyamines in cells have become an attractive pathway for therapeutic intervention on polyamine depletion therapy[22–28]. For example, Li and coworkers developed a peptide-pillar [5] arene conjugate to trap polyamines in tumor cells for suppressing tumor growth through host–guest encapsulation[22]. Even though these methods have succeeded to a certain degree, there are still some challenges for clinical applications[29]. The key limitation of these approaches is that cells have the capacity to take up polyamines from the systemic circulation to compensate for the drug-induced loss of polyamines. The catching of polyamines using host–guest interaction within the cells also faces similar issues. In addition, the caught polyamines may be released back to the cells as the host–guest interaction is reversible. To overcome these disadvantages, developing a novel, irreversible polyamine depletion strategy for cancer therapy is urgently required.

Herein, we proposed and explored a new concept for polyamine detection and possible depletion by designing a new photosensitizer that can react with polyamines chemically so the overall polyamines can be depleted. In addition, the chemical reaction of this photosensitizer with polyamines generates two photosensitizers that can produce more reactive oxygen species (ROS) for synergistic cancer destruction. Therefore, this combination of polyamine consumption via chemical reaction can provide a new modality to enable polyamine detection along with photodynamic therapy as well as a depletion of polyamines for cancer treatment and prevention as illustrated in Fig. 1.

## Results

The concept is based on pyrrolopyrrole aza-BODIPY pyridinium salt (**PPAB-PyS**) nanoparticles that can react with the overexpressed polyamine in cancer cells and produce two photosensitizers (**DPP-PyS** and compound **3**) with strong ROS generation capability for more effective cancer cell destruction. As shown in Fig. 1, **PPAB-PyS** is composed of three parts: a planar conjugated PPAB core, two electron-deficient pyridinium units, and a propeller-shaped triphenylethylene (TPE) unit with a PPAB core at one end. According to our previous report[30–32], the PPAB core can react with polyamine effectively and generate two less conjugated reaction products. The TPE part endows compound **3** with aggregation-induced emission (AIE) properties with bright fluorescence and an improved ROS production. In addition, the two pyridinium units increase the water solubility of **PPAB-PyS**, which is important for biomedical applications.

As depicted in Supplementary Fig. 1, according to our previous methods[30–32], **PPAB-PyS** was prepared from a compound **1** with two heteroaromatic amines (compounds **2** and **3**), followed by the Suzuki reaction and a nucleophilic reaction step by step. Its chemical structure was characterized by NMR spectroscopy and HRMS (Supplementary Figs. 1 and 2). **PPAB-PyS** has three absorption peaks at ~ 670, 619, and 430 nm and emission from 710 to 723 nm in different solvents, as shown in Supplementary Fig. 3a, b. Density functional theory calculation indicates that the electron densities of the HOMO and LUMOs of **PPAB-PyS** are located on the PPAB core and cationic pyridinium moiety, respectively. The energy gap ($E_g$) is 1.656 eV (Supplementary Fig. 3c).

The reactivity of **PPAB-PyS** with polyamines can be observed by its respective reaction with putrescine, spermidine, and spermine, as shown in Fig. 2. For spermine-pretreated **PPAB-PyS**, the absorption peaks at 670, 619, and 430 nm rapidly disappeared and the peak at 327 nm red-shifted to 350 nm, implying the formation of new species with less conjugation in this reaction. After 30 s, there are no spectral changes, indicating the reaction reached equilibrium. Meanwhile, its emission peak at 710 nm was quickly decreased while the emission at 430 nm slightly increased upon the addition of spermine. The 20-fold enhancement of the $I_{430}/I_{710}$ ratio was observed (Fig. 2). The reaction with spermidine orputrescine is basically the same as that with spermine. The reaction with other primary amines (such as ethanediamine, diaminopropane, cadaverine, n-propylamine, n-butylamine, n-hexylamine, and cyclohexylamine), secondary amine (diethylamine), tertiary amines (triethylamine, trimethylamine), and aromatic amines (phenylamine) were also tested as demonstrated in Supplementary Figs. 4 and 5. All of them show similar behaviors as reflected in the changes in the UV–vis and emissions. The ratios of $A_{350}/A_{670}$ and $I_{430}/I_{710}$ for different amines were summarized in Supplementary Table 2. Based on our observations, the reactivity in terms of reaction rate is in the following order: polyamines > primary aliphatic amines > secondary amines » tertiary amines and aromatic amines. These trends can be further confirmed by the calculated pseudo-first-order rate constants ($k_{obs}$), as summarized in Supplementary Fig. 6 and Table 3, respectively. As discussed earlier[32], since the first step reaction is rate-determining and kinetically controlled, polyamine and diamine could more easily destroy aza-BODIPY rings of **PPAB-PyS** through cleavage of B–N bond than monoamine. Then, it is possible that the second amino group intramolecularly attacks the remaining B–N bond, which is much easier and faster than that of an intermolecular attack by another amine. Subsequently, hydrolysis could be undergone to generate final reaction products. So, the more amino groups in amine, the faster is the reaction rate.

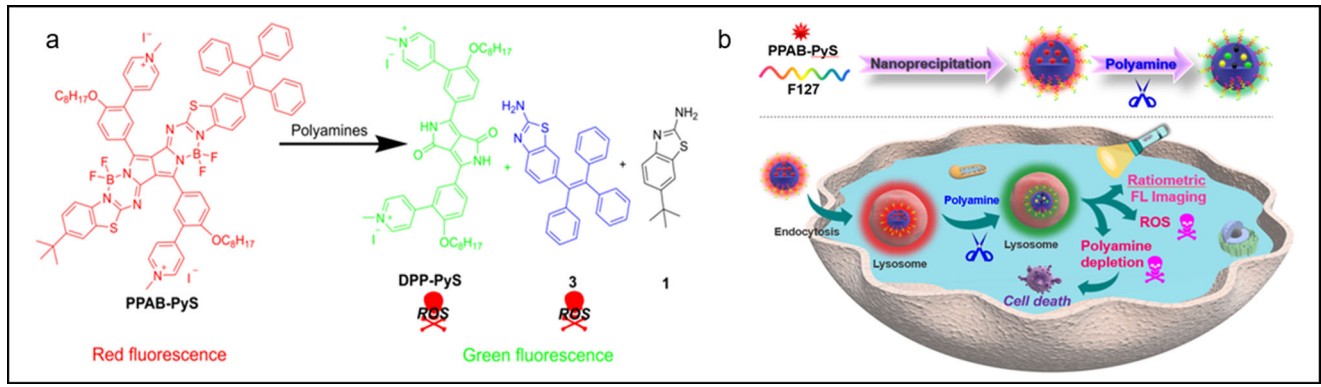

**Fig. 1 The combination of photodynamic therapy with a putative depletion of polyamines on cancer treatment. a** The proposed reaction between **PPAB-PyS** and polyamines. **b** Schematic illustration of lysosome-targeting, polyamine depletion, enhanced ROS generation, and ratiometric fluorescence imaging for the cancer cells.

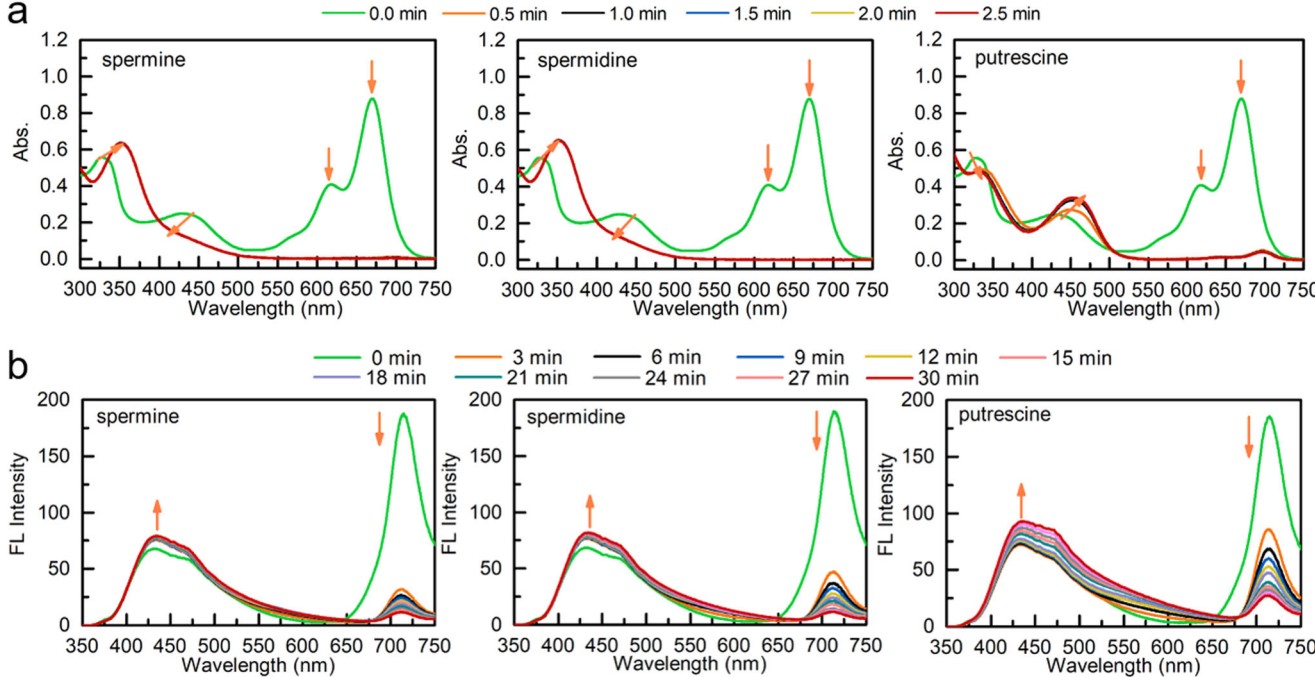

**Fig. 2 The reactivity of PPAB-PyS with polyamines.** Time-dependent **a** UV–vis and **b** emission spectra of PPAB-PyS (10 µM) in MeCN in presence of spermine spermidine putrescine at 25 °C.

Moreover, as shown in Supplementary Fig. 7, **PPAB-PyS** cannot react with other biological species (GSH, $H_2O_2$, metal ions, and anions) under the same conditions. **PPAB-PyS** is also stable in acidic conditions (Supplementary Fig. 8). These results indicate **PPAB-PyS** has a specific high reaction selectivity to polyamines.

We also evaluated the polyamine concentration-dependent performance of **PPAB-PyS** for the quantitative detection of polyamines. As shown in Supplementary Fig. 9, there is an excellent linear relationship between the decrease of absorption at 670 nm with the polyamine concentration, suggesting the potential application of **PPAB-PyS** for quantitative polyamine detection. The detection limits are found to be 0.193, 0.326, and 0.602 µM for spermine, spermidine, and putrescine, respectively. This sensitivity makes it possible to detect intracellular polyamines.

The reaction mechanism between **PPAB-PyS** and polyamine (putrescine as a model compound) was investigated with HRMS and NMR spectrometry. The HRMS spectra in Supplementary Fig. 10 indicated the final products (**DPP-PyS** and two hetero-aromatic amines **2** and **3**) were present by the peaks at 726.1 for [**DPP-PyS** −2H]+, 191.1 for [**2** + H]+, 402.5 for [**3** − 2H]+. Two new BF2-chelating complexes (**BF1** and **BF2**) were also observed, in which the peaks at 226.5 and 137.2 are assigned to [**BF1**]+ and [**BF2** + H]+, respectively. The [1]H[11], B, and [19]F NMR spectra further confirmed the formation of these products (Supplementary Figs. 11–13). The appearance signal at 11.1 and 11.2 ppm in [1]H NMR spectrum indicated the presence of proton signal for N–H of **DPP-PyS**. The original triplet at 1.30 ppm up-shifted to 0.89 ppm and a broad peak at 0.06 ppm was present in [11]B NMR spectrum. The broad peak at −130.45 ppm for **PPAB-PyS** disappeared, but multiple broad peaks at −129.28 to −129.71, −133.20 to −135.10 ppm appeared in [19]F NMR spectrum. The changes of [11]B and [19]F spectra indicated that new B–F and B–N species different from that of **PPAB-PyS** were formed. Based on

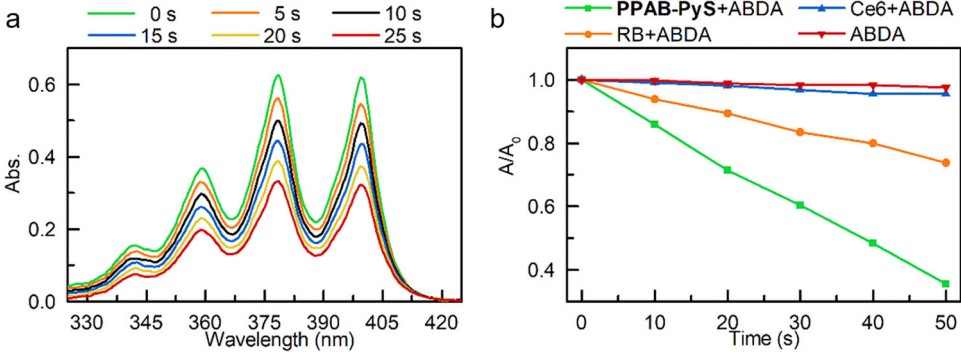

**Fig. 3 The generation of singlet oxygen by PPAB-PyS. a** UV–vis spectra of ABDA with **PPAB-PyS** under white light irradiation in MeCN/water mixtures with $f_w = 99\%$. **b** Plot of relative absorbance of ABDA without and with different PSs under white light irradiation, where $A_0$ and $A$ are the absorbances of ABDA at 378 nm before and after white light irradiation, respectively. $[PS] = 5 \times 10^{-6}$ M, $[ABDA] = 5.0 \times 10^{-5}$ M.

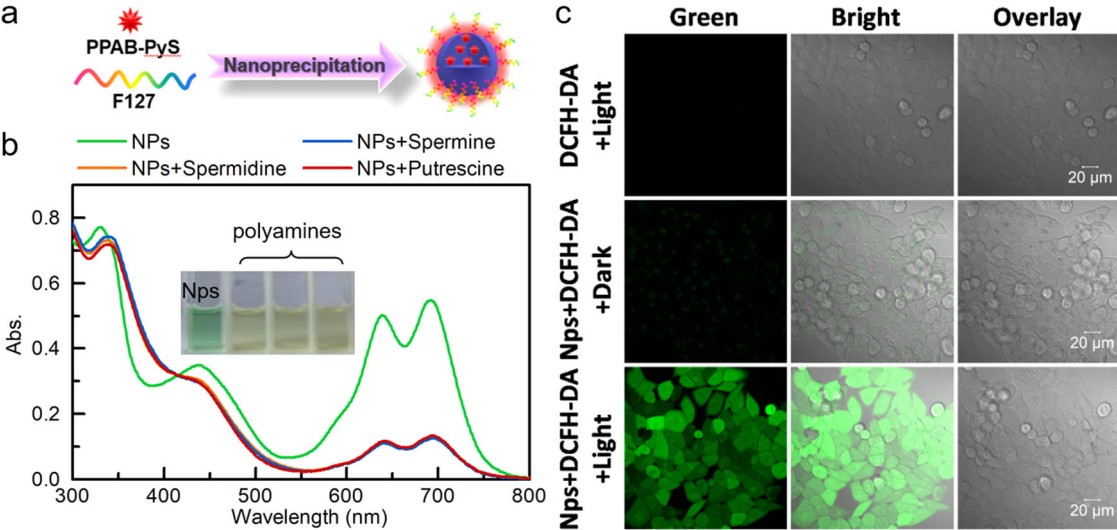

**Fig. 4 The detection of intracellular ROS generated by PPAB-PyS NPs. a** The formation of **PPAB-PyS** NPs. **b** The UV–vis spectra of **PPAB-PyS** NPs in presence of polyamines (inset: the color change in a presence of polyamines). **c** Intracellular ROS detection using DCFH-DA (10 μM, 30 min) as an indicator in HeLa cells after **PPAB-PyS** NPs (50 μM) under white light (30 mW/cm²) irradiation for 30 min.

these observations, the proposed reaction mechanism was shown in Supplementary Fig. 2.

By use of 9,10-anthracenediyl-bis(methylene)-dimalonic acid (ABDA) as a $^1O_2$ probe, the $^1O_2$ generation capability of **PPAB-PyS** was investigated under white light irradiation. As shown in Fig. 3, the absorbance of ABDA decreased sharply in the presence of **PPAB-PyS**, which was more distinct than that of commercially Chlorin e6 (Ce6) and Rose Bengal (RB). After 50 s exposure to white light, compared to more than 60% ABDA consumption for **PPAB-PyS**, only 5 and 25% ABDA degraded for Ce6 and RB, respectively. The results reveal that **PPAB-PyS** is more efficient than Ce6 and RB for $^1O_2$ generation. As shown in Supplementary Fig. S14, taking RB as the standard PS (0.75 in water), the $^1O_2$ quantum yield of **PPAB-PyS** was calculated to be 87.5%.

**PPAB-PyS** was encapsulated into pluronic F-127 to form nanoparticles in order to improve their biocompatibility and water solubility as well as to facilitate intracellular uptake. The hydrodynamic size of **PPAB-PyS** NPs was found to be 57.4 nm with a low polydispersity index (0.12) as measured by dynamic light scattering (DLS). It was found that the size did not display any changes over 4 days in water, revealing their excellent stability (Supplementary Fig. 15).

The reactivity of **PPAB-PyS** NPs with mM putrescine, spermidine, and spermine were studied systematically. As shown in

Supplementary Fig. 16, the size and polydispersity index of **PPAB-PyS** NPs were increased to 87.1 (0.29), 89.2 (0.37), and 91.3 nm (0.40) when 0.40 mM putrescine, spermidine, and spermine were added, respectively. **PPAB-PyS** NPs showed the same reaction performances when reacted with polyamines as that of **PPAB-PyS** solution (Fig. 3a), generating distinct ratiometric color change.

The detection of intracellular ROS was performed under white light irradiation with 2′,7′-dichlorodihydrofluorescein diacetate (H2DCF-DA), an intracellular ROS indicator. As shown in Fig. 4b, when HeLa cells were treated with **PPAB-PyS** NPs and H2DCF-DA, bright green emission was observed. On the contrary, there was nearly no emission in the control groups (**PPAB-PyS** NPs + H2DCF-DA in the dark and H2DCF-DA + light). Therefore, **PPAB-PyS** NPs generated ROS in HeLa cells, under light irradiation.

As polyamines are over-expressed in some tumors, we particularly explored whether **PPAB-PyS** NPs could react with the endogenous polyamine in cancer cells as a potential solution for cancer treatment as shown in Figs. 5 and 6, respectively. The imaging performance of **PPAB-PyS** NPs in HeLa cells for different incubation durations was tracked. As shown in Fig. 5a, **PPAB-PyS** NPs showed red emission. After incubation for 4 h, the emission in the green channel was clearly illuminated, likely ascribed to the less-conjugated products (**DPP-PyS** and two

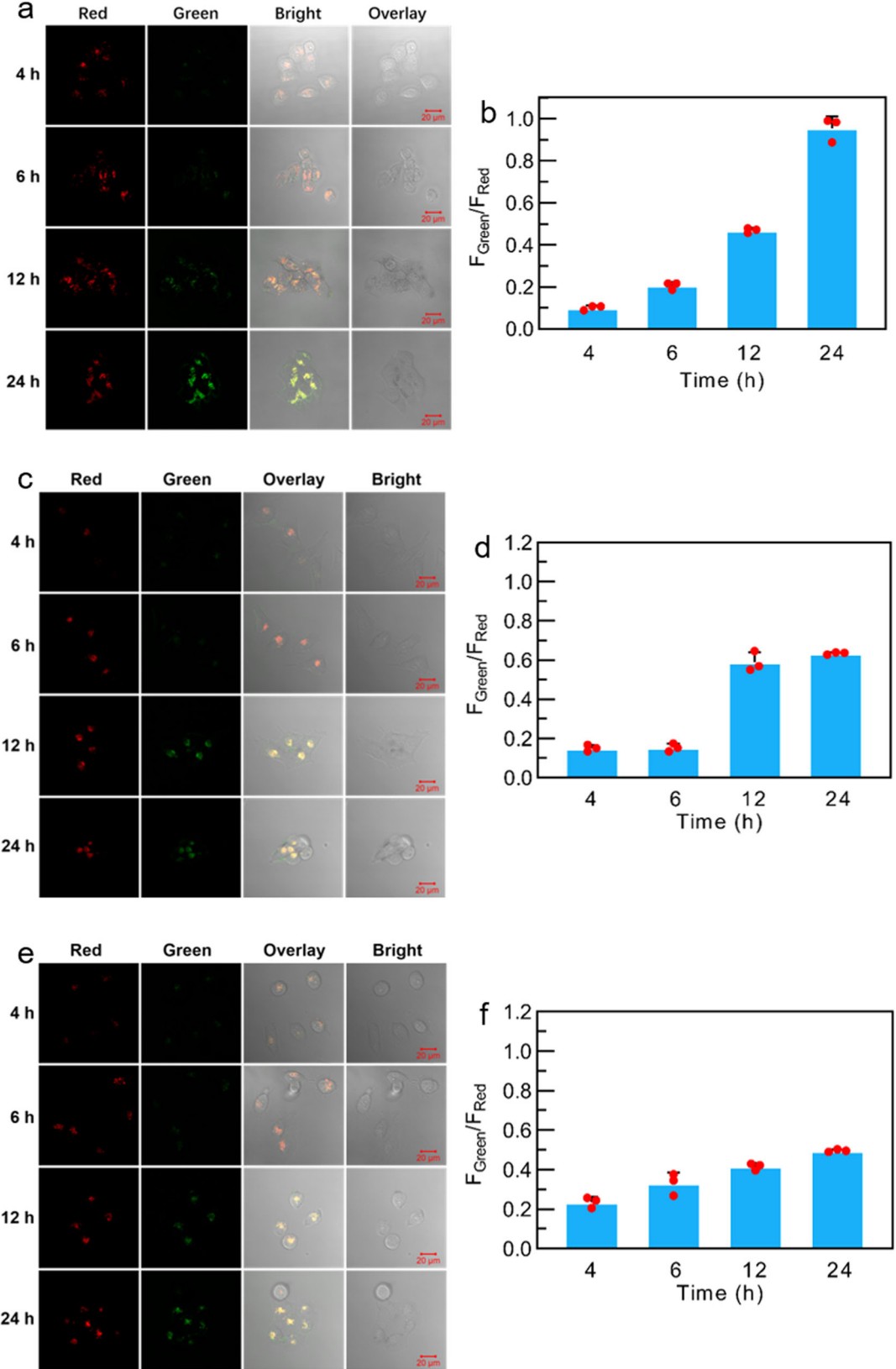

**Fig. 5 Time-dependent ratiometric fluorescence bioimaging.** CLSM imaging and $F_{green}/F_{red}$ data of **a**, **b** HeLa, **c**, **d** MCF-7, and **e**, **f** LO2 cells incubated with **PPAB-PyS** NPs (100 μM) for 4, 6, 12, and 24 h.

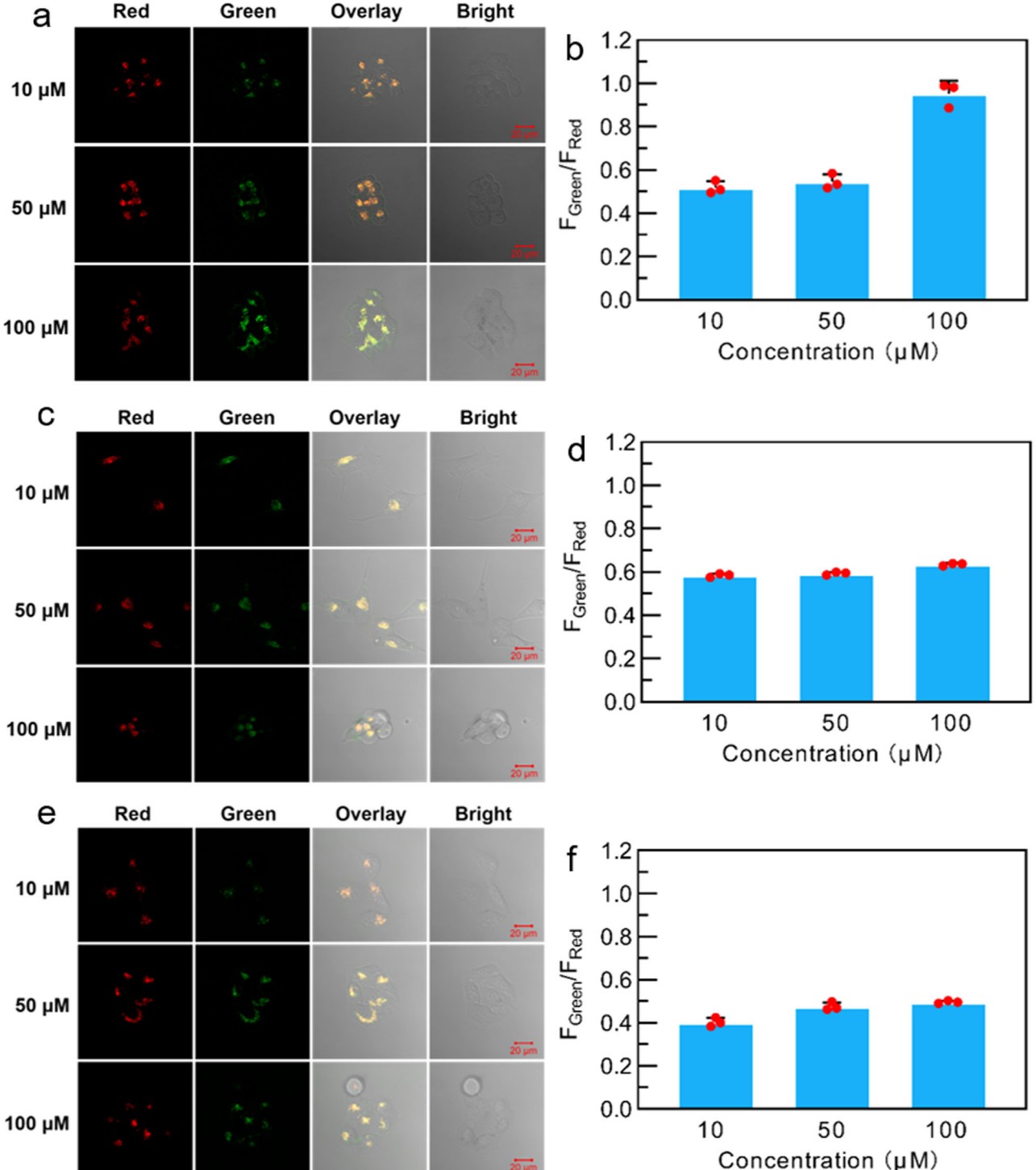

**Fig. 6 Concentration-dependent ratiometric fluorescence bioimaging.** CLSM images and $F_{green}/F_{red}$ values of (**a**, **b**) HeLa, (**c**, **d**) MCF-7, and (**e**, **f**) cells stained with **PPAB-PyS** NPs (10, 50, 100 μM) for 24 h.

heteroaromatic amines **1** and **2**) produced from the chemical reaction between **PPAB-PyS** NPs and the endogenous polyamines. With the increasing incubation duration from 4 to 24 h, the green emission intensity in HeLa cells was enhanced gradually and the ratios of $F_{green}/F_{red}$ were increased from 0.1 to 1. Furthermore, the microscopic images and the in situ emission spectra of HeLa cells stained with **PPAB-PyS** NPs for 24 h indicated that the red and green emission maximum was located at 555 and 700 nm, respectively (Fig. 7). These results indicated that **PPAB-PyS**

NPs reacted with endogenous polyamines to produce less-conjugated products. A similar ratiometric fluorescent imaging was observed for MCF-7 cancer cells and LO2 normal cells but with lower $F_{green}/F_{red}$ values (Fig. 5b, c). For instance, the $F_{green}/F_{red}$ values were 0.6 and 0.5 after 24 h incubation time. The ratios of $F_{green}/F_{red}$ changed in this order: HeLa cells > MCF-7 cells > LO2 cells.

Meanwhile, the concentration-dependent ratiometric fluorescence bioimaging of HeLa, MCF-7, and LO2 cells under CLSM

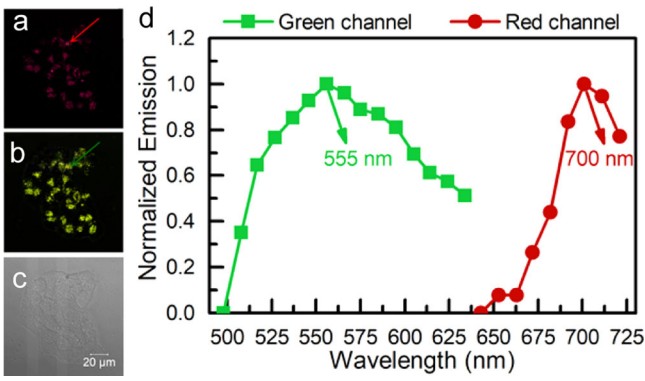

**Fig. 7 The microscopic images and the in situ emission spectra. a** Red and **b** Green **c** Bright channel of real color images of live HeLa cells stained with **PPAB-PyS** NPs (100 μM) for 24 h. **d** In situ emission spectra presented with green and red color were acquired in the circular are as in (**a, b**) images. Red channel, $\lambda_{ex} = 633$ nm, $\lambda_{em} = 643–721$ nm; Green channel, $\lambda_{ex} = 488$ nm, $\lambda_{em} = 498–634$ nm.

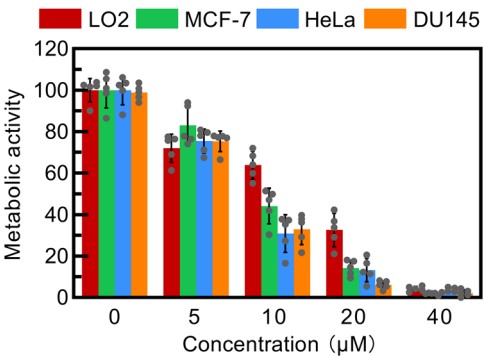

**Fig. 8 The phototoxicity of PPAB-PyS NPs.** Cell metabolic activity of HeLa, MCF-7, DU145, and LO2 cells stained with different concentrations of **PPAB-PyS** NPs under white light for 30 min.

was investigated. As shown in Fig. 6, when **PPAB-PyS** NPs concentration was increased, the emission in the red channel began to weaken and the emission in the green channel became stronger for all cell lines. For a particular example in HeLa cells, the $F_{green}/F_{red}$ showed the most distinctive concentration-dependent effect, where the $F_{green}/F_{red}$ values increased from 0.51 to 0.96 with the increase of **PPAB-PyS** NPs concentration from 10 to 100 μM. MCF-7 and LO2 cells displayed modest $F_{green}/F_{red}$ changes from 0.59 to 0.63, and 0.40 to 0.50 under the same conditions, respectively. These results further demonstrate that ratiometric fluorescent imaging toward endogenous polyamine was successfully achieved.

Inspired by the remarkable intracellular ROS generation of **PPAB-PyS** and ratiometric fluorescence imaging of endogenous polyamines, in vitro phototoxicity was investigated in HeLa cells via MTT assays. It is noted that the intrinsic absorptions of **PPAB-PyS** NPs in the absence and presence of polyamines are not overlapped with that of formazan absorbance, MTT can provide an accurate evaluation of the cell metabolic activities. After exposure to white light irradiation, the cell viability of HeLa cells significantly decreased with the increase of the concentration of **PPAB-PyS** NPs and prolonged incubation time (Supplementary Fig. 17). Upon white light irradiation for 30 min, dose-dependent cytotoxicity was observed. For instance, the cell viability of 40%, 10%, and 5% was shown when **PPAB-PyS** NPs concentration was increased from 10, 50, and 100 μM,

respectively. In presence of 100 μM **PPAB-PyS** NPs, the cell metabolic activity significantly decreased to 35%, 10%, and 5% with prolonging the irradiation time of 10, 20, 30 min, respectively. So, the **PPAB-PyS** NPs concentration below 50 μM and 30 min irradiation were selected as following photocytotoxicity study.

The photocytotoxicity of **PPAB-PyS** NPs in different cancer cells (HeLa, MCF-7, DU145 cancer cells, and LO2 cells as a control) was further examined. A shown in Fig. 8, **PPAB-PyS** NPs induced the best dose-dependent cytotoxic effect against HeLa cells under white light irradiation. Cell metabolic activity of HeLa cells in presence of **PPAB-PyS** NPs (40 μM) was reduced to 5% after 30 min irradiation, indicative of an excellent potential PDT capability (Fig. 8). **PPAB-PyS** NPs also displayed considerable anticancer bioactivity against MCF-7 and DU145 cells. Compared to non-malignant LO2 cells, **PPAB-PyS** NPs showed much higher toxicity against malignant cell lines. In addition, the IC$_{50}$ values of 7.71, 9.62, 8.27, and 16.20 μM were determined for HeLa, MCF-7, DU145, and LO2 cell lines, respectively. More importantly, as shown in Supplementary Table 4, **PPAB-PyS** NPs displayed better anticancer performance than a supramolecular trap for polyamine sequestration, where up to tenfold lower IC$_{50}$ value was obtained.

The live- and dead-cell staining experiments (where live and dead cells were stained by calcein AM (green fluorescence) and propidium iodide (red fluorescence), respectively) are used to further justify the selective anticancer efficacy of **PPAB-PyS** NPs. As shown in Fig. 9a, when HeLa cells were treated with **PPAB-PyS** NPs under white light irradiation for 30 min, the intense red fluorescence and very weak green fluorescence were observed, which is consistent with the results from MTT assays. Comparatively, cells in the control groups showed no red emission but strong green emission. These results confirm the high cytotoxicity of **PPAB-PyS** NPs against HeLa cells.

In order to reveal the mechanism behind the antitumor activity of **PPAB-PyS** NPs, some control experiments were carried out. Firstly, the ROS generation capability of reaction products (**2, 3,** and **DPP-PyS**) was evaluated. As shown in Supplementary Fig. 18, **3** is a typical AIE-active molecule, where the emission intensity dramatically increased with the increase of water content. As shown in Supplementary Fig. 19, both **3** and **1** (the analog of **DPP-PyS**) can efficiently generate ROS as indicated by H2DCF-DA fluorescence, where green emission intensity gradually strengthened with prolonged irradiation time. On the contrary, **2** failed to generate ROS under the same conditions. More importantly, the combination of **1** and **3** exhibited better ROS generation capability than **PPAB-PyS** (Fig. 9b), which is expected to enhance the PDT efficacy.

## Discussion

It is well-known that polyamine depletion would disturb normal physiological effects and excessive depletion of polyamines would cause cell death. As discussed above, the irreversible chemical reaction between **PPAB-PyS** NPs and the endogenous polyamines would facilitate polyamine-deficiency to cause cell death. To evaluate the effect of polyamine depletion on cell viability, cancer cells were incubated with **PPAB-PyS** NPs in dark. As shown in Fig. 8c, only slight cytotoxic activity was observed for LO2 regular cells. The cell viability decreased from 100 to 92% for LO2 cells when **PPAB-PyS** NPs concentration increased from 5 to 40 μM. However, the cell viability decreased from 100 to 50% for DU145 cells under the same test conditions. PPAB-PyS NPs have a better anticancer ability against DU145 cells than LO2 cells. This is because PPAB-PyS NPs are more effective in the consumption of polyamines. The prostate is the organ in the body

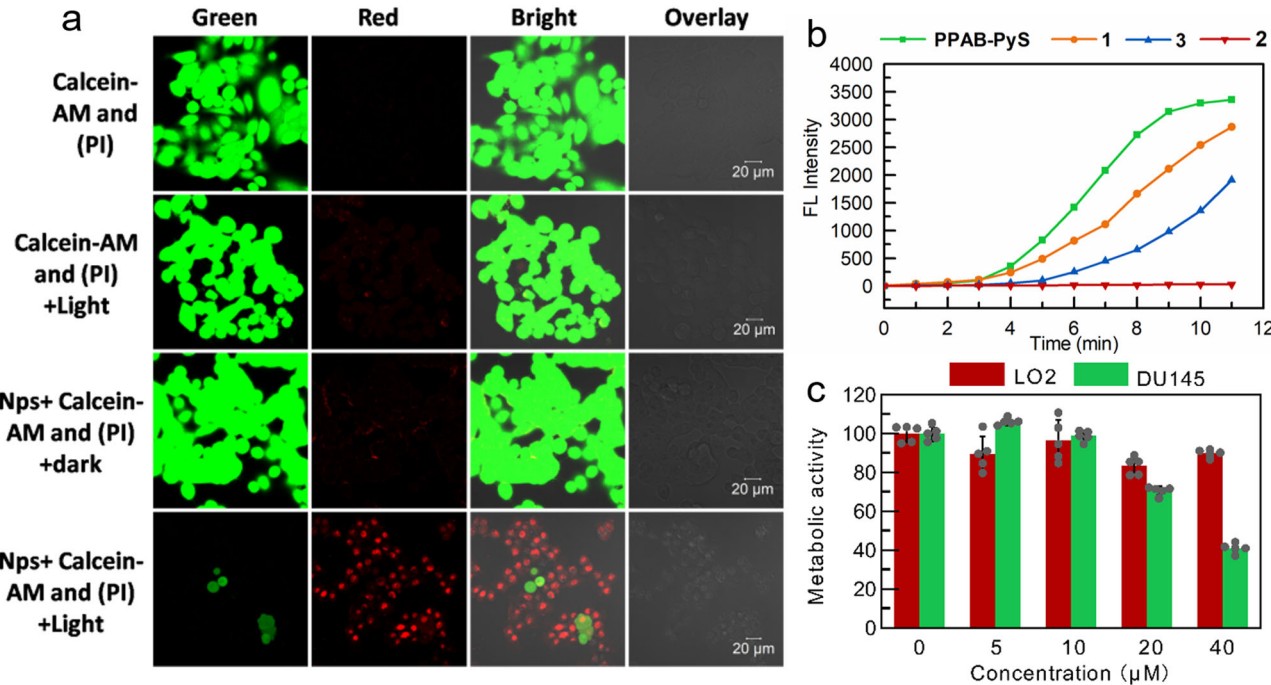

**Fig. 9 The live- and dead-cell staining observations. a** CLSM images of HeLa cells stained with **PPAB-PyS**NPs (50 μM) incubated HeLa cells co-stained by Calcein-AM and PI after irradiation (30 mW/cm², 30 min). **b** The PL intensity at 540 nm of H2DCF-DA in the presence of **PPAB-PyS**,**1**,**2** and **3** upon white-light irradiation for different times. Concentration: $10 \times 10^{-6}$ M (**PPAB-PyS, 1, 2**, and **3**), $2 \times 10^{-4}$ M (H2DCF-DA); $\lambda_{ex}$: 488 nm. **c** Cell viability of DU145 and LO2 cells stained with different concentrations of **PPAB-PyS** NPs in dark.

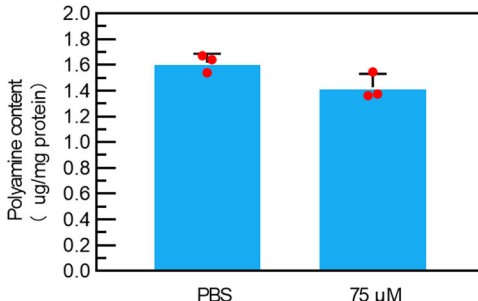

**Fig. 10 The spermine determination by HPLC.** HPLC analysis was used to detect the intracellular spermine concentration in DU145 cells ($n = 3$, mean ± SD).

that contains the highest polyamine content in the human body, and therefore, prostate cancer is closely related to polyamine development[33]. DU145 cells are more sensitive to polyamine consumption than LO2 cells, this also indicates that the better efficiency of **PPAB-Pys** NPs on DU145 cells than on LO2 cells is most likely from polyamine depletion.

According to previous studies[23–25], spermine is the most abundant and representative polyamine in cancer cells. Based on a previous method[22], the spermine determination was conducted by the HPLC method. In this study, we found **PPAB-PyS** NPs induce a decrease of intracellular spermine in DU145 cells (Fig. 10). The preliminary results suggest a putative polyamine depletion as the deduction on the polyamine is so significant as expected.

Nanomaterials are attractive in lysosome targeting because most NPs can finally reach lysosomes as a result of the lysosomal membrane fusion. In addition, lysosomal microenvironments are favorable for the ROS-induced apoptosis pathway. Programmed ROS generation in lysosomes of cancer cells is important for

enhanced cancer treatment[34–36]. To further investigate the therapeutic mechanism of **PPAB-PyS** NPs, the subcellular co-localization experiment was carried out with an organelle-specific fluorescent dye LysoTracker® Green. It is interesting to find that **PPAB-PyS** NPs can specifically target lysosome (Fig. S20). The perfect overlapping of **PPAB-PyS** NPs and lysosome gave rise to a high overlap coefficiency (0.91), indicating **PPAB-PyS** NPs are lysosome-targeting photosensitizers. So, **PPAB-PyS** NPs accumulated in the lysosome and emitted red fluorescence after endocytosis. When **PPAB-PyS** NPs reacted with the endogenous polyamine, the corresponding products with a green emission were also found to target lysosome with a high overlap coefficiency (0.81), as shown in Supplementary Fig. S21. The lysosome-targeting **PPAB-PyS** NPs are beneficial to initiate cancer cell death through lysosome-specific damage.

We reckon that several factors have accounted for the highly efficient anticancer activity of **PPAB-PyS** NPs. First, the selective lysosome-targeting capability of **PPAB-PyS** NPs is important, because ROS-induced lysosomal damage would lead to the release of lysosomal cathepsins into the cell cytoplasm, subsequently causing the initiation of cell death. Second, **PPAB-PyS** NPs can efficiently react with the endogenous polyamine in tumor cells, resulting in irreversible polyamine consumption, which has a negative effect on the physiological processes of cancer cells. Finally, under irradiation, the gradual ROS generation (from **3** and **DPP-PyS**) during polyamines depletion in cancer cell lysosomes is another important effect for ROS-induced apoptosis and enhanced cancer treatment. Therefore, the combined effects of lysosome-targeting capability, gradual ROS generation, and simultaneous consumption of polyamine likely work synergistically for the improved anticancer activity of **PPAB-PyS** NPs.In addition, the consumption of polyamine in the bloodstream via the reaction with **PPAB-PyS** nanoparticles may also provide an effective method for cancer prevention as the reduction of poly-amine may prevent cancer cell development.

Most cancers have high polyamine levels. For example, colorectal cancer shows increased three to fourfold polyamine contents over those found in the equivalent normal tissue. Some groups revealed depletion of the intracellular polyamine pools invariably inhibited cell growth[31]. The effect of depletion of polyamines on growth inhibition in mammalian cells through various pathways such as a total arrest in mitochondria-mediated apoptosis, protein translation, or cell cycle arrest has been investigated[5]. Here, our findings showed that the consumption of polyamines induced by **PPAB-PyS** NPs plus PDT resulted in greatly decreased cell proliferation in cancer cells.

In summary, we described a new concept of polyamine consumption cancer therapy based on novel **PPAB-PyS** NPs that can react with, and consume the over-expressed polyamine in cancer cells and produce two photosensitizers with enhanced ROS capability for a synergistic cancer cell destruction. We have successfully demonstrated that lysosome targeting **PPAB-PyS** NPs are appropriate agents for fluorescence-guided polyamine depletion therapy via irreversible chemical reactions. More importantly, ROS generation from the reaction products of **3** and **DPP-PyS**, further improves the anticancer efficacy of the system. This study offers an irreversible polyamine consumption strategy, in combination with PDT, to boost the anticancer efficacy of this unique treatment modality along with the potential for cancer prevention. However, we must point out that the consumption inside the cells was not sufficient and the depletion is mainly putative. The reason is most likely DU145 cells rebalance their polyamine pools via biosynthesis, importing exogenous polyamines, interconverting their existing pools, etc. Since cells have the capacity to take up polyamines from the systemic circulation to compensate for the drug-induced loss of polyamines, the decrease of polyamine by **PPAB-PyS** nanoparticles can be recovered when cells were cultured in high-glucose Dulbecco's Modified Eagle's Medium (H-DMEM) containing 10% fetal bovine serum (FBS). Because the channels for polyamine are so many as pointed out above, their depletion is a challenging task. Our work is just at a very early stage and much work is needed to be done. We think the combination of inhibition of polyamine metabolism (difluoromethylornithine, DFMO), targeting polyamine transport system(polyamine analogs or polyamine conjugates) with our **PPAB-PyS** nanoparticles will have a better result, which we will conduct in the following project.

## Methods

All chemicals and reagents were purchased from commercial sources and used as received without further purification[1]. H NMR[11], B NMR, and $^{19}$F NMR spectra were carried on a Bruker spectrometer. The UV–vis absorption spectra were recorded using a Helios Alpha UV-Vis scanning spectrophotometer with a 1 cm quartz cell. Fluorescence spectra were quantitatively measured by a FluoroMax-4 spectrofluorometer with a xenon lamp and 1 cm quartz cells. Confocal laser scanning microscope (CLSM) characterization was conducted with a confocal laser scanning biologicalmicroscope (LSM 710, Zeiss, Germany). The absorbance for MTT analysis was recorded on a microplate reader (Thermo Fisher, USA) at a wavelength of 570 nm. Size and zeta potential measurements were conducted on DLS (ZSE, Malvern, UK).

**Synthesis and characterization of compounds**. The synthetic route for compound **PPAB-PyS** is shown in Fig. 1. **PPAB-Py** was synthesized according to our previous method[31]. Compound **PPAB-Py** (60 mg, 0.046 mmol), methyl iodide (49 mg, 0.184 mmol) was added to a 25 mL two necked, round-bottom flask, then 10 mL toluene were injected into the flask. The flask was evacuated under vacuum and flushed with dry nitrogen three times. After the reaction was further refluxed for 3 h, the solution was filtrated. The residue was washed with toluene. **PPAB-PyS** was obtained as dark green solid in 73% yield[1]. H NMR (CDCl$_3$, 600 MHz, ppm): $\delta$ = 9.44–9.42 (d, $J$ = 12, 2H), 9.31–9.30 (d, $J$ = 6, 2H),8.64–8.59 (m, 5H), 8.41–8.36 (m, 4H), 7.74–7.72 (d, $J$ = 12, 1H), 7.64 (s, 1H), 7.75–7.49 (m, 2H), 7.69–7.18 (m, 18H), 4.74 (s, 1H), 4.42–4.19 (m, 4H), 1.98–1.82 (m, 4H), 1.47–1.29 (m, 29H), 0.92–0.88 (m, 6H). HRMS ($m/z$):$m/z$ calcd. for C$_{84}$H$_{84}$B$_2$F$_4$N$_8$O$_2$S:$^{2+}$ 699.3161, found: 699.3147.

**Preparation of PPAB-PySNPs**. A modified co-precipitation method was used to prepare **PPAB-PyS** NPs. In a typical experiment, 2 mg of **PPAB-PyS** dissolved in 1 mL THF solution, and 5 mg of pluronic F-127 was dissolved in 5 mL deionized water 1 mL, and the **PPAB-PyS** solution was injected into pluronic F-127 solution. Then the mixture was evaporated to remove THF completely on a rotary evaporator at 40 °C. The resulting mixture was cooled to room temperature. The obtained solution of **PPAB-PyS** NPs was filtered by a 0.22 μm filter to discard big aggregates and then stored at 4 °C for further use.

### General experimental operation

*ROS detection.* 2,7-Dichlorodi-hydrofluorescein diacetate (H2DCF-DA, 200 μM) is used as a ROS indicator. **PPAB-PyS**, **1**, **2**, and **3** were prepared as 10 μM in MeCN. Spermine was prepared as 800 μM in H$_2$O. Then the cuvette was exposed to white light (300–700 nm) for different amounts of time, and the fluorescence spectra were observed immediately after each irradiation. In the control group, an H2DCF-DA solution without PSs was subjected to irradiation. The fluorescence was excited at 488 nm.

**$^1$O$_2$ quantum yield measurements via the chemical method**. 9,10-Anthracene-diyl-bis(methylene)-dimalonic acid (ABDA) was used as the $^1$O$_2$ indicator, and Rose Bengal (RB) was employed as the standard photosensitizer. To eliminate the inner-filter effect, the absorption maximum as adjusted to ~0.2 OD. The measurements were carried out under white light irradiation in DMSO/water mixtures with $f_w$ = 99%. [ABDA] = 5 × [PSs], ROS quantum yields of **PPAB-PyS** were calculated by the equation: $\Phi_{PPAB-PyS} = \Phi_{RB}*(K_{PPAB-PyS}/K_{RB})*(A_{RB}/A_{PPAB-PyS})$, where $K_{PPAB-PyS}$ and $K_{RB}$ are the decomposition rate constants of ABDA with **PPAB-PyS** and RB, respectively. A$_{PPAB-PyS}$ and A$_{RB}$ represent the light absorbed by **PPAB-PyS** and RB, respectively, which are determined by the integration of the areas under the absorption bands in the wavelength range of 300–800 nm. $\Phi_{RB}$ is the $^1$O$_2$ quantum yield of RB, which is 0.75 in water[37].

**Cell culture**. Cells were cultured in high-glucose Dulbecco's Modified Eagle's Medium (H-DMEM) containing 10% FBS and 1% penicillin-streptomycin at 37 °C in a humidified environment containing 5% CO$_2$. Before the experiment, the cells were pre-cultured until confluence was reached.

**Cell imaging**. HeLa, MCF-7, DU145, and LO2 cells were seeded in the 12-well plate and cultured in H-DMEM with 10% FBS at 37°C in a humidified environment containing 5% CO$_2$. After an 80% confluence, the medium was removed and the adhered cells were rinsed twice with 1× PBS. Cells were incubated with different concentrations of **PPAB-PyS** NPs for different times. After washing the culture dishes three times with PBS, fluorescence imaging experiments were carried out on an LSM710 confocal microscope (Carl Zeiss, Germany).

**Detection of Intracellular ROS**. HeLa cells were incubated with **PPAB-PyS** NPs (50 μM) for 24 h. Then, cells were washed with 1× PBS for three times, and fresh medium containing 10 μM of DCFH-DA (Sigma-Aldrich) was added. After further incubation for 30 min, the cells were washed three times with 1× PBS and then irradiated by white light (35 mW cm$^{-2}$, 30 min). Finally, the fluorescence images of the cells were taken by a CLSM. The fluorescence of DCF was excited at 488 nm and collected within 500–560 nm.

**Cytotoxicity assay by MTT**. The cytotoxicity of **PPAB-PyS** NPs was examined by Cell Counting MTT method. HeLa, MCF-7, DU145, and LO2 cells were grown in 96-well plates with a confluence of about $5.0 \times 10^3$ cells/well. Then, 100 μL fresh culture medium containing different concentrations of **PPAB-PyS** NPs were added into different cell plates. After incubation for 24 h at 37 °C, the old cell culture medium was discarded, and the cells were washed twice or more with PBS and further incubated with 100 μL fresh medium containing 5 mg mL$^{-1}$ of MTT at 37 °C for another 4 h. Then the medium was removed and 100 μL DMSO was added. Finally, the absorbance at 570 nm was measured by a microplate reader. Each experiment was run in triplicate. The equation: Cell viability (%) = ($A_{probe}$ − $A_{blank}$)/($A_{control}$ − $A_{blank}$) × 100% was used to calculate the cell viability.

**Cytotoxicity of photosensitizers to different cells under light irradiation**. The cells were seeded in 96-well plates at a density of 5000 cells per well. After 24 h at 37 °C, the medium was replaced with a fresh medium (100 μL) containing different concentrations of **PPAB-PyS**. After incubation for 12 or 24 h, the cells were exposed to white light (30 mW/cm$^2$) for 30 min, and another array of plates with the cells containing different concentrations of **PPAB-PyS** were kept in the dark as a control. After that, the plates were subjected to the same treatment as the cytotoxicity study.

**Live/dead cell imaging of PPAB-PyS NPs-treated HeLa cells**. HeLa cells with 80% confluence were incubated with **PPAB-PyS** NPs for 24 h, the cells were irradiated by white light (30 mW/cm$^2$, 30 min). Then the cells were incubated with Calcein-AM (10 μg mL$^{-1}$) and propidium iodide (3 μg mL$^{-1}$) for 30 min.

Afterward, cells were washed and taken for confocal imaging with an excitation at 488 nm (for Calcein-AM) and 543 nm (for propidiumiodide).

**Preparation of isolated/cultured cells.** DU145 cells were seeded into plates at $1 \times 10^6$ cells/well and cultured for 24 h under 5% $CO_2$ at 37 °C. Then the serum-containing medium was removed, and cells were rinsed twice with PBS. Further cells were cultured with a serum-free medium in the absence or presence of **PPAB-PyS** NPs for 24 h. After treatment, the cells were harvested by trypsinization and then were resuspended in PBS. A part of the cell suspension was used to determine the protein content according to the BCA method, and another part of the suspension was added with a perchloric acid solution to obtain cell extractive.

**Spermine determination by HPLC.** The amount of spermine was analyzed by high-performance chromatography in DU145 in the absence or presence of **PPAB-PyS** NPs. Briefly, the cells were harvested in 200 μL of PBS, and 200 μL of hexanediamine (1.624 μg/ml) was added as an internal standard. After adding 200 μL of dansyl chloride (5 mg/ml), the mixtures were incubated at 50 °C for 30 min, and then the reaction was terminated by the addition of 1 mL of ethyl acetate. The supernatant containing the polyamines was functionalized with dansyl chloride and purified with an organic filtration. Totally, 20 μL of the sample was then injected onto an XDB-C18 column (4.6 × 250 mm, Agilent Technologies), which was achieved excitation at 340 nm and measured emission at 515 nm by a fluorescence detector. The solvent system was consisted of methanol and water, running at 65% (v/v) to 100% (v/v) methanol within 25 min at a flow rate of 1 mL/min. BCA assay kit was used to determine the concentration of protein.

**Statistics and reproducibility.** Data in this study are represented as mean ± standard deviation of three replicate experiments. All error bars shown are the standard error of the mean.

**Reporting summary.** Further information on research design is available in the Nature Research Reporting Summary linked to this article.

## Data availability

The data that support the findings of this study are available from the authors on reasonable request, see author contributions for specific data sets.

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

## Acknowledgements

The supports by the China Scholarship Council (201906155012), the National Natural Science Foundation of China (22071065 and 21772045), the Natural Science Foundation of Guangdong Province (2018B030311008), and the Technology Program of Guangzhou (201904010414) are gratefully acknowledged. The authors are also grateful to the University of Macau (Grant no. MYRG2019-00059-ICMS) for partial financial support of

this work. WC would like to acknowledge the partial support from Solgro, Guangxi Jialuoyuan Biomedical Inc., and the distinguished award from UT Arlington.

## Author contributions

L.Y.W, R.B.W., and W.C. conceived the project, designed the experiments, and wrote the paper. W.T.L., T.L.S., and H.T. completed the experiments. B.B. and D.C. helped with data measurements and data analysis. All authors discussed and commented on the paper.

## Competing interests

The authors declare no competing interests.

## Additional information

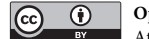

