## [Transparent Peer Review File · Communications Biology]

Reviewers' comments:

Reviewer #1 (Remarks to the Author):

The authors are presenting a new take on polyamine depletion cancer therapy based on PPAB-PyS nanoparticles that simultaneously provide intracellular imaging and ratiometric detection of the intracellular polyamine. The photosensitized the irreversible chemical reaction between PPAB-PyS NPs and endogenous polyamines allows for the depletion of those and simultaneous cancer cell death. This method is novel as it is based on multifunctional nanoparticles that allow for both controlled treatment and detection and the polyamines depletion pathway is underexplored. Nanoparticles are well-characterized and their effects are studied in several cell lines indicating cell line/polyamine overexpression dependence. The internalization of these nanoparticles is well assessed indicating desired lysosomal colocalization for further lysosome-specific treatment. The study is of high impact as it offers cancer-specific controlled non-toxic to healthy cells treatment alternative. I suggest accepting the article for publication with a few minor suggestions/corrections:

- Line 110, it should be selectivity
- Line 160: reduced instead of illuminated? Likely ascribed should be separated
- In the MTT assays it would be highly beneficial for authors to describe if the intrinsic absorbance of the nanoparticles in the region of the formazan absorbance can or does interfere with the assay.
- Line 190 further examined should be separated
- Line 224 regular instead of cancer
- Line 227: separate against and DU145
- Font in the synthesis and characterization section has to be adjusted

Reviewer #2 (Remarks to the Author):

The manuscript (COMMSBIO-20-2834-T) entitled, "Combination Of Chemical Reaction For Depleting Polyamines With Photodynamic Therapy As A Novel Modality For Cancer Treatment And Prevention" by Li et al describes the development of an amine-reactive fluorescent molecule which degrades into smaller molecules which are active in photodynamic therapy. There are several interesting features of this paper. From the organic chemistry perspective, this is really a paper about boron-amine complexation exchange and the subsequent hydrolysis of the original boron ligand to form a conjugated array which photodynamically generates ROS. From the polyamine perspective, the rate differences between different polyamine and amine constructs is interesting especially the observation that spermidine has the fastest rate of reaction rate in its reaction with the lead molecule PPAB-PyS as implied in Supplementary Table 2 (note this table's title is missing 'PPAB-PyS'), presumably because it has an available aminobutyl group. Note putrescine was faster than 1,3-diaminopropane, which is consistent with this conclusion. From the biology perspective, the fact that this PPAB-PyS construct forms three separate products upon reaction with polyamines two of which are photoreactive makes understanding its biological activity complex due to three different factors in play. The authors go on to show uptake into mammalian cells and involvement of lysosomes. So in this regard, the paper may be interesting to a wide audience. However, there are numerous issues with the data and data interpretation that suggest this needs more work before it is worthy of Nature.

First, the Supplementary Fig 10, shows a complex mass spectral pattern with specific signals assigned to a specific molecule (fragment of the original). These products need to be isolated and separately characterized to confirm their identity as was done in Supp Fig 13 albeit with NMR, but they need to show the respective mass spectrum and UV VIS spectrum of each purified byproduct too (compounds 1, 2 and 3).

The authors incorrectly state that MTT measures viability. It does not. It measures metabolic activity in the conversion of a formazan derivative. The authors are likely measuring changes in cell growth

not cell viability here. If they wish to discuss viability they need a different assay like their propidium iodide study. Also they need to show that the presence of the PPAB-PyS agent alone without cells does not interfere with the MTT formazan signal measured by the plate reader. This is an important control as the presence of their photo-reactive agent may interfere or compete with this readout.

There are many biological amines that can compete for their boron laden probe and these amines could potentially liberate the sequestered polyamines from the suggested polyamine-boron complex. For example, in an organic laboratory, methanol is typically added to boron-amine complexes and the volatility of the trimethoxyborane allows for boron removal. This is why knowledge of the rates of formation is key to provide some insight into the affinity of the different amine constructs (as the authors have provided). Since spermidine has the fastest on rate the authors should independently make the borane spermidine complex and test its stability to other biological amines. Does the spermidine come off or is it stuck in the borane complex? Is this truly polyamine depletion or just formation of a reversible polyamine pool that is sometimes bound to boron? What is the off rate for spermidine (Spd) from the Spd-boron complex?

There is no description of the statistical analyses performed on the data as far as I could tell.

I do laud the authors for their creative approach to sequester polyamines but feel that this manuscript has too many holes to be accepted in its current form. Perhaps the weakest aspect of the paper is the claim of polyamine depletion without actually measuring intracellular polyamine pools by LCMS or N-dansylation and HPLC. For this reason alone, the paper should be rejected. To assist the authors in their future efforts to publish their work, I am providing my personal edits of their manuscript.

Reviewer #3 (Remarks to the Author):

Chen and Wang et al. reported the combination of a chemical reaction for depleting polyamines and photodynamic therapy as a novel antitumor strategy. The results indicated that PPAB-PyS nanoparticles can effectively consume polyamines via an irreversible chemical reaction. More importantly, the resulting products can generate ROS under light irradiation for synergistic cancer treatment. Due to the novelty of the concept as well as the convincing experimental evidence, I would like to recommend this manuscript for publication by Communication Biology after a minor revision to address the following questions:

1. The previously reported, polyamines-related cancer therapeutic agents, such as DFMO, polyamine analogues and polyamine conjugates, and supramolecular traps, can be summarized in a Table in the Supporting Information for facile comparison.
2. What is the reason about "the reactivity in terms of reaction rate is in the following order: polyamines > primary aliphatic amines > secondary amines >> tertiary amines and aromatic amines."? Please give a brief explanation in the manuscript, with appropriate citations.
3. The Figure S17 should be shown in the main text, because it is important to show the reaction between PPAB-PyS NPs and endogenous polyamine.

Minor issues:

1. Overall the manuscript is well written, but there are still occasional grammatical mistakes. The language of the manuscript should be carefully examined again before acceptance.
2. The small tick-marks in Figure 4, Figure 5, and Figure 6c should be deleted.

Responses to Referee's comments

Reviewer 1

The authors are presenting a new take on polyamine depletion cancer therapy based on **PPAB-PyS** nanoparticles that simultaneously provide intracellular imaging and ratiometric detection of the intracellular polyamine. The photosensitized the irreversible chemical reaction between **PPAB-PyS** NPs and endogenous polyamines allows for the depletion of those and simultaneous cancer cell death. This method is novel as it is based on multifunctional nanoparticles that allow for both controlled treatment and detection and the polyamines depletion pathway is underexplored. Nanoparticles are well-characterized and their effects are studied in several cell lines indicating cell line/polyamine overexpression dependence. The internalization of these nanoparticles is well assessed indicating desired lysosomal colocalization for further lysosome-specific treatment. The study is of high impact as it offers cancer-specific controlled non-toxic to healthy cells treatment alternative. I suggest accepting the article for publication with a few minor suggestions/corrections:

1 Line 110, it should be selectivity.

Response: Thank you, we changed it.

2 Line 160: reduced instead of illuminated? Likely ascribed should be separated

Response: Yes, we changed them.

3 In the MTT assays it would be highly beneficial for authors to describe if the intrinsic absorbance of the nanoparticles in the region of the formazan absorbance can or does interfere with the assay.

Response: This is a good question. For polyamine-pretreated **PPAB-PyS** NPs, the absorption peaks at 670, 619 and 430 nm rapidly disappeared and the peak at 327 nm red-shifted to 350 nm, as shown in Figure 3. According to Wang's report (Journal of Microbiological Methods, 2010, 82, 330-33), the absorption spectrum of formazan showed a peak between 510 and 570 nm with a single absorption maximum at approximately 550 nm. In our experiments, the absorption at 550 nm (A_{550}) was chosen for monitoring the formazan production. There is no overlap with MTT's absorptions. Therefore, the intrinsic absorptions of the **PPAB-PyS** nanoparticles in the region of the formazan absorption is not interfered with that of the assay. We have added these explanations in the revised paper.

Fig. 3. The formation of **PPAB-PyS** NPs and their UV-Vis spectra in presence of polyamine.

4 Line 190 further examined should be separated.

Response: Corrected

5 Line 224 regular instead of cancer.

Response: Corrected.

6 Line 227: separate against and DU145.

Response: Corrected

7 Font in the synthesis and characterization section has to be adjusted.

Response: Adjusted

Reviewer 2

The manuscript (COMMSBIO-20-2834-T) entitled, “Combination Of Chemical Reaction For Depleting Polyamines With Photodynamic Therapy As A Novel Modality For Cancer Treatment And Prevention” by Li et al describes the development of an amine-reactive fluorescent molecule which degrades into smaller molecules which are active in photodynamic therapy. There are several interesting features of this paper. From the organic chemistry perspective, this is really a paper about boron-amine complexation exchange and the subsequent hydrolysis of the original boron ligand to form a conjugated array which photodynamically generates ROS. From the polyamine perspective, the rate differences between different polyamine and amine constructs is interesting especially the observation that spermidine has the fastest rate of reaction rate in its reaction with the lead molecule **PPAB-PyS** as implied in Supplementary Table 2 (note this table’s title is missing ‘**PPAB-PyS**’), presumably because it has an available aminobutyl group. Note putrescine was faster than 1,3-diaminopropane, which is consistent with this conclusion. From the biology perspective, the fact that

this **PPAB-PyS** construct forms three separate products upon reaction with polyamines two of which are photoreactive makes understanding its biological activity complex due to three different factors in play. The authors go on to show uptake into mammalian cells and involvement of lysosomes. So in this regard, the paper may be interesting to a wide audience. However, there are numerous issues with the data and data interpretation that suggest this needs more work before it is worthy of Nature. 1 The Supplementary Fig 10, shows a complex mass spectral pattern with specific signals assigned to a specific molecule (fragment of the original). These products need to be isolated and separately characterized to confirm their identity as was done in Supp Fig 13 albeit with NMR, but they need to show the respective mass spectrum and UV VIS spectrum of each purified byproduct too (compounds 1, 2 and 3).

Response: Thank you! According to the reaction mechanism, **PPAB-PyS** would generate **DPP-PyS** pyridinium salt (**DPP-PyS**) and two aromatic amines (**2** and **3**) in presence of polyamine. Due to the strong polarity of the three compounds, it's difficult to separate them by a common column chromatography. As shown in Figure S1, since the three compounds have a strong interaction with silica gel, the TLC silica gel plate showed a serious tailing phenomenon. We could not isolate them. Instead, we used HRMS, ¹H, ¹¹B, ¹⁹F NMR data to confirm the reaction mechanism.

Figure S1 The photographs of TLC silica gel plate of **PPAB-PyS** in the absence and presence of spermine under daylight and 365 nm UV light.

2 The authors incorrectly state that MTT measures viability. It does not. It measures metabolic activity in the conversion of a formazan derivative. The authors are likely measuring changes in cell growth not cell viability here. If they wish to discuss viability they need a different assay like their propidium iodide study. Also they need to show that the presence of the **PPAB-PyS** agent alone without cells does not interfere with the MTT formazan signal measured by the plate reader. This is an important control as the presence of their photo-reactive agent may interfere or compete with this readout.

Response:

MTT (3-(4,5-dimethylthiazol-2-yl)-2,5-diphenyltetrazolium bromide) tetrazolium assay is a popular tool in estimating the metabolic activity of living cells. The test is based on the enzymatic reduction of the lightly colored tetrazolium salt to its formazan of intense purple-blue color, which can be quantified spectrophotometrically. Under properly optimized conditions the obtained absorbance value is directly proportional to the number of living cells [S1-S3].

Cell viability testing, as opposed to measuring the metabolic activity of viable cells, requires evaluation at the level of single cells or discrete groups of cells. The use of flow cytometry which can now be adapted to a microplate format. Recently, digital imaging microscopy methods have also been applied to cell viability testing using dyes like trypan blue that are excluded from viable cells, but they can enter and bind to proteins when the integrity of the plasma membrane is compromised. Dyes that enter cells and generate a fluorescent signal following binding to DNA (e.g., propidium iodide) and proteins are also used to measure cell viability. Tetrazolium dyes, however, are not ideal reagents for measuring the percentage of viable cells because their formazans are either crystalline which can itself damage cell membranes.

Based on your comments and the above discussions, we have changed the expression and discussion of MTT in text and Figures. The “viability” has been changed to “metabolic activity”.

PPAB-PyS has three absorption peaks at ~670, 619 and 430 nm, which are not overlapped with that of formazan. The presence of the **PPAB-PyS** agent alone without cells does not interfere with the MTT formazan signals.

[S1] M.V. Berridge, M.P. Herst, A.S. Tan, Tetrazolium dyes as tools in cell biology: new insights into their cellular reduction. *Biotech. Annu. Rev.*, 11 (2005), 127-152.

[S2] Ewa Grela, Joanna Kozłowska, Agnieszka Grabowiecka, Current methodology of MTT assay in bacteria – A review, *Acta Histochemica*, 2018, 120, 303-311.

[S3] Razmik Mirzayans, Bonnie Andrais, David Murray, Viability assessment following anticancer treatment requires single-cell visualization, *Cancers* 2018, 10, 255; doi:10.3390/cancers10080255.

3 There are many biological amines that can compete for their boron laden probe and these amines could potentially liberate the sequestered polyamines from the suggested polyamine-boron complex. For example, in an organic laboratory, methanol is typically added to boron-amine complexes and the volatility of the trimethoxyborane allows for boron removal. This is why knowledge of the rates of formation is key to provide some insight into the affinity of the different amine constructs (as the authors have provided). Since spermidine has the fastest on rate the authors should independently make the borane spermidine complex and test its stability to other biological amines. Does the spermidine come off or is it stuck in the borane complex? Is this truly polyamine depletion or just formation of a reversible polyamine pool that is sometimes bound to boron? What is the off rate for spermidine (Spd) from the Spd-boron complex? There is no description of the statistical analyses performed on the data as far as I could tell.

Response:

Yes, these are very good points. According to our previous results (*Sensors and Actuators B: Chemical*, 2020, 312, 127953; *Chemical Communications*, 2019, 55, 9789-9792), we think the interactions between **PPAB-PyS** and polyamines starts with the first kinetic-control reaction of a B-N bond cleavage by polyamines following by a fast hydrolysis reaction to yield much smaller conjugated molecules (Scheme S2). The first kinetic-control reaction generates borane-polyamine complex, which is found in MS spectra in Figure S10. Moreover, ^{11}B and ^{19}F NMR data (Figures S11 and S12) indicated that that new B-F and B-N species different from that of **PPAB-PyS** were formed.

From the intracellular spermine concentration in DU145 cells HPLC analysis in Supplementary Figure 22, it is obvious that spermine content is decreased. It seems that spermine come off in the borane complex. These preliminary observations support the concept reported in this work.

Supplementary Scheme S2. The possible reaction mechanism between **PPAB-PyS** and putrescine.

4 I do laud the authors for their creative approach to sequester polyamines but feel that this manuscript has too many holes to be accepted in its current form. Perhaps the weakest aspect of the paper is the claim of polyamine depletion without actually measuring intracellular polyamine pools by LCMS or N-dansylation and HPLC. For this reason alone, the paper should be rejected. To assist the authors in their future efforts to publish their work, I am providing my personal edits of their manuscript.

Response: Thank you for your positive comments. Our work is just at the beginning, much work in in progress to find more evidence to support our hypothesis.

Polyamines are essential for normal cell growth and differentiation, and aberrant polyamine metabolism is known to play an important role in the development of tumors [S4-S5]. According to previous studies (*Chem. Commun.* 2019, 55, 2340-2343, *ACS Appl. Mater. Interfaces* 2018,10, 5365-5372, *ACS Appl. Mater. Interfaces*, 2017, 9, 8602-8608), spermine is the most abundant and representative polyamine in cancer cells. Based on the previous method (*Nat. Commun.* 2019,10, 3546), the polyamine determination by HPLC. In this study, we found **PPAB-PyS** NPs induce decrease of intracellular spermine in DU145 cells (Supplementary Figure 22). The preliminary results support the claim of polyamine depletion therapy and are consistent with the CLSM imaging and $F_{\text{green}}/F_{\text{red}}$ data in Figure 4 and Figure 5.

The polyamine determination by HPLC was added in the Supporting Information.

Supplementary Figure 22 HPLC analysis was used to detect the intracellular spermine concentration in DU145 cells (n =3, mean \pm SD).

[S4] Berrak O, Akkoc Y, Arisan ED, Coker-Gurkan A, Obakan-Yerlikaya P, Palavan-Unsal N. The inhibition of PI3K and NFkappaB promoted curcumin-induced cell cycle arrest at G2/M via altering polyamine metabolism in Bcl-2 overexpressing MCF-7 breast cancer cells. *Biomed Pharmacother.* 2016; 77:150-60.

[S5] Igarashi K, Kashiwagi K. Modulation of cellular function by polyamines. *Int J Biochem Cell Biol.* 2010; 42: 39-51.

Reviewer 3

Chen and Wang et al. reported the combination of a chemical reaction for depleting polyamines and photodynamic therapy as a novel antitumor strategy. The results indicated that **PPAB-PyS** nanoparticles can effectively consume polyamines via an irreversible chemical reaction. More importantly, the resulting products can generate ROS under light irradiation for synergistic cancer treatment. Due to the novelty of the concept as well as the convincing experimental evidence, I would like to recommend this manuscript for publication by *Communication Biology* after a minor revision to address the following questions:

1 The previously reported, polyamines-related cancer therapeutic agents, such as DFMO, polyamine analogues and polyamine conjugates, and supramolecular traps, can be summarized in a Table in the Supporting Information for facile comparison.

Response: We have provided a Supplementary Table 1 in the Supporting Information.

Supplementary Table 1. Inhibitors that target polyamine metabolism, function and transport.

Inhibitors	Target	Structure	Status	Ref
DFMO	ODC		Approved for the treatment of Trypanosoma brucei subsp gambiense ; Multiple ongoing clinical trials for cancer, including prostate cancer, lung cancer, and colon cancer.	1,2
BENSpm	SSAT or SMO		Preclinical use	3
CHENSpm	SSAT		Preclinical use	4
F14512	DNA		Clinical trials for the treatment of refractory/relapsing acute myeloid leukemia	5
PAP5A			Experiments in vivo show that P1P5A effectively inhibits the growth of breast adenocarcinoma xenografts in female nude mice	6

2 What is the reason about “the reactivity in terms of reaction rate is in the following order: polyamines > primary aliphatic amines > secondary amines >> tertiary amines and aromatic amines.”?

Please give a brief explanation in the manuscript, with appropriate citations.

Response: According to our previous results (Sensors and Actuators B: Chemical, 2020, 312, 127953), the reactivity in terms of reaction rate for different amines can be ascribed to as follows. As discussed earlier, since the first step reaction is rate-determining and kinetically controlled, polyamine and diamine could more easily destroy *aza*-BODIPY rings of **PPAB-PyS** through cleavage of B-N bond than monoamine. Then, it is possible the second amino group intramolecularly attacks the remaining B-N bond, which was much easier and faster than that of intermolecular attack by another amine. Subsequently, the hydrolysis could be undergone to generate final reaction

products. So, more amino groups in amine have, faster reaction rate involves. We have added the discussion in the revised text.

3 The Figure S17 should be shown in the main text, because it is important to show the reaction between **PPAB-PyS** NPs and endogenous polyamine.

Response: We have moved Figure S17 to the text as Fig.6.

Minor issues:

1. Overall the manuscript is well written, but there are still occasional grammatical mistakes. The language of the manuscript should be carefully examined again before acceptance.

Response: We have carefully revised the grammatical mistakes, spelling mistakes and others in whole manuscript.

2. The small tick-marks in Figure 4, Figure 5, and Figure 6c should be deleted.

Response: We have edited Figure 4, Figure 5, and Figure 6c according to reviewer's comment.

Reviewers' comments:

Reviewer #2 (Remarks to the Author):

Polyamine depletion is a challenging task because cancer cells have many ways to rebalance their polyamine pools. They can make more via biosynthesis, they can import exogenous polyamines or they can interconvert their existing pools if a particular need arises. The authors need much stronger data to make the claim of this system as a polyamine depletion agent in vitro.

While I appreciate the authors attempts to address my concerns, I am still not satisfied with the data showing these molecules actually reduce polyamine levels in living cells. The Supporting Information Figure 22, shows 94.4% of the initial spermine level still remaining in DU145 cells treated with 75 uM of the compound! This is within the error of the N-dansylation analysis involving an extractive workup and does not show spermine depletion as the authors suggest. Note: this type of data is typically shown as nmol polyamine/mg protein (not as ug/mg as the authors show). The fact that the y-axis of Fig S22 was expanded to skew the visual to look like a significant decrease in this figure was not appreciated (See 'How to Lie with Statistics' by Darrel Huff). If the authors want to publish this as a detection system for polyamines then the data is supportive of that, but not as a polyamine depletion agent. As such I would suggest rejection of this paper in its current format as it does not prove what it claims.

Reviewer #3 (Remarks to the Author):

I have read the revised paper and the response to the reviewers, and I think my comments have been addressed. In my opinion, it can be published in Commun Bio. I also saw the question about the Figure S22 and noticed the doubts from reviewer 2. I think that the results are normal because the spermine synthase can produce the spermine when it is depleted. The polyamines should be consumed because they are reactants in this system. If the authors investigated the levels of spermine synthases, spermidine synthase and ornithine decarboxylase in the cells with and without the effective compounds by WB, the results can possibly prove whether the polyamines are depleted. Specially, the ornithine decarboxylase was more active than spermine and spermidine synthase.

Responses to reviewer's comments

Reviewer #2 (Remarks to the Author):

Polyamine depletion is a challenging task because cancer cells have many ways to rebalance their polyamine pools. They can make more via biosynthesis, they can import exogenous polyamines or they can interconvert their existing pools if a particular need arises. The authors need much stronger data to make the claim of this system as a polyamine depletion agent in vitro.

While I appreciate the authors attempts to address my concerns, I am still not satisfied with the data showing these molecules actually reduce polyamine levels in living cells. The Supporting Information Figure 22, shows 94.4% of the initial spermine level still remaining in DU145 cells treated with 75 μ M of the compound! This is within the error of the N-dansylation analysis involving an extractive workup and does not show spermine depletion as the authors suggest. Note: this type of data is typically shown as nmol polyamine/mg protein (not as μ g/mg as the authors show). The fact that the y-axis of Fig S22 was expanded to skew the visual to look like a significant decrease in this figure was not appreciated (See 'How to Lie with Statistics' by Darrel Huff). If the authors want to publish this as a detection system for polyamines then the data is supportive of that, but not as a polyamine depletion agent. As such I would suggest rejection of this paper in its current format as it does not prove what it claims.

Response: We appreciate the valuable comments and suggestions from the reviewer. The reviewer is right that the Supporting Information Figure 22 shows 94.4% of the initial spermine level still remaining in DU145 cells treated with 75 μ M of the compound. Even if the intracellular spermine concentration in DU145 cells by HPLC analysis showed that change in absence and presence of PPAB-PyS nanoparticles is not very big but 5.6 % is obviously beyond the experimental errors. We appreciate that the reviewer mentioned 'How to Lie with Statistics' by Darrel Huff which is a nice book for investigators like us. To correct that, we changed the Y-axis in Fig. 22 from 1.45 – 1.65 to 0.00 – 1.65. However, this change did not affect the result that the change is 5.6 % of the intracellular spermine concentration in DU145 cells in absence and presence of PPAB-PyS nanoparticles. The change of 5.6 % corresponding to 8.96 ng which is far beyond the HPLC error of 0.01 ng.

In this work, we provided three methods to prove that the overall polyamines were depleted:

- 1) The reaction of PPAB-Pys with polyamine to form DPP-Pys and two heteroaromatic amines **1** and **2** by high resolution Mass Spectroscopy, NMR and luminescence. The experiments provide

many evidences to support the chemical reaction of PPAB-Pys with polyamines.

- 2) The Absorption and fluorescence measurements prove that both in the nutrients and inside the cells, the polyamines were consumed and depleted. The imaging performance of **PPAB-PyS** NPs in HeLa cells for different incubation durations was tracked. As shown in Fig. 4a, **PPAB-PyS** NPs showed red emission. After incubation for 4 h, the emission in green channel was clearly illuminated, likely ascribed to the less-conjugated products (**DPP-PyS** and two heteroaromatic amines **1** and **2**) produced from the chemical reaction between **PPAB-PyS** NPs and the endogenous polyamines. With the increasing incubation duration from 4 to 24 h, the green emission intensity in HeLa cells was enhanced gradually and the ratios of $F_{\text{green}}/F_{\text{red}}$ were increased from 0.1 to 1. Furthermore, the microscopic images and the *in-situ* emission spectra of HeLa cells stained with **PPAB-PyS** NPs for 24 h indicated that the red and green emission maximum were located at 555 and 700 nm, respectively (Fig. 5). These results indicated that **PPAB-PyS** NPs reacted with the endogenous polyamines to produce less-conjugated products.
- 3) The HPLC as discussed here.

We agreed with the reviewer that the depletion inside the cells was not sufficient. Just as pointed by the reviewer that it is most likely DU145 cells rebalance their polyamine pools via biosynthesis, importing exogenous polyamines, interconverting their existing pools, *etc.* Since cells have the capacity to take up polyamines from systemic circulation to compensate for drug induced loss of polyamines, the decrease of polyamine by PPAB-PyS nanoparticles can be recovered when cells were cultured in high-glucose Dulbecco's Modified Eagle's Medium(H-DMEM) containing 10% fetal bovine serum (FBS). As reviewer #2 proposed, polyamine depletion is a challenging task. Our work is just at a very early stage and much work is needed to be done. We think the combination of inhibition of polyamine metabolism (difluoromethylornithine, DFMO), targeting polyamine transport system (polyamine analogues or polyamine conjugates) with our PPAB-PyS nanoparticles will have a better result, which we will conduct in the following project.

Based on the recommendation of the reviewer and the kind suggestions from the editor, we would like to tone the depletion down in some levels and focus on the polyamine detection and photodynamic therapy. So, we changed the title to '**Combination of chemical reaction for**

polyamine detection with photodynamic therapy as well as a possible depletion of polyamine on cancer treatment and prevention’. We hope this is acceptable for the reviewer and the editors.

The Figure 22 in Supporting Information has been adjusted according to standard visual method. The type of data is typically shown as nmol polyamine/mg protein. However, ug/mg is also acceptable, quite many papers used ug/mg (*Nat. Commun.* 2019, 10, 3546; *Oncotarget*, 2017, 8, 1092). For the raw data we got, ug/mg is more convenient, so the Y-axis of Fig S22 was shown in ug polyamine/mg protein.

Reviewer #3 (Remarks to the Author):

I have read the revised paper and the response to the reviewers, and I think my comments have been addressed. In my opinion, it can be published in *Commun Bio*. I also saw the question about the Figure S22 and noticed the doubts from reviewer 2. I think that the results are normal because the spermine synthase can produce the spermine when it is depleted. The polyamines should be consumed because they are reactants in this system. If the authors investigated the levels of spermine synthases, spermidine synthase and ornithine decarboxylase in the cells with and without the effective compounds by WB, the results can possibly prove whether the polyamines are depleted. Specially, the ornithine decarboxylase was more active than spermine and spermidine synthase.

Response: Thank you for your positive comments!

We appreciate your advice to investigate the levels of spermine synthases, spermidine synthase and ornithine decarboxylase in the cells with and without the effective compounds by WB. However, our method is not to suppress the biosynthesis of polyamines by the cancer cells. So, we do not think it is really helpful to the levels of spermine synthases, spermidine synthase and ornithine decarboxylase in the cells with and without the effective compounds by WB. We think the combination of inhibition of polyamine metabolism (difluoromethylornithine, DFMO), targeting polyamine transport system (polyamine analogues or polyamine conjugates) with our PPAB-PyS nanoparticles would be more effective, which we will conduct in the following project.

Reviewers' comments:

Reviewer #1 (Remarks to the Author):

The changes made by the author are superficial and did not address my concern about polyamine depletion. A 5% decrease in polyamine pools is not considered significant. If one conducts the entire experiment from start to finish three times and averages the results, a standard deviation is encountered as much as 5-10% error, which is what the authors claim as a significant depletion effect. It is not. Perhaps they could conduct the proposed DFMO combination experiment to demonstrate true cellular polyamine depletion and show a synergistic effect with DFMO and their agent. Otherwise, I suggest they drop the claim of polyamine depletion altogether and focus on polyamine detection aspects of their method.

Reviewer #2 (Remarks to the Author):

To whom it may concern:

I have independently evaluated the manuscript entitled, "Combination of chemical reaction for polyamine detection with photodynamic therapy as well as a depletion of polyamine on cancer treatment and prevention", by Li et al. In the abstract the authors claim, "... new photosensitizer ispyrrolopyrroleaza-BODIPY pyridinium salt 20 (PPAB-PyS) nanoparticles that can react with the over-expressed polyamine in cancer cells and produce two photosensitizers with enhanced phototoxicity on cancer destruction. Meanwhile, PPAB-PyS nanoparticles provide a simultaneous ratio metric fluorescence imaging of intracellular polyamine... This combination ... as well as a depletion of polyamines for cancer treatment and prevention"

Reviewer2 questions the depletion aspect. He/she states, "I am still not satisfied with the data showing these molecules actually reduce polyamine levels in living cells."

In my view, the main figures in a publication must directly support the claims made in the abstract. Two out of three of the reviewers are apparently satisfied, but reviewer2 strongly objects to the third claim "that the use of the nanoparticle photosensitizing agent depletes polyamines". Fig. 1 does support this claim shown with model amine compounds (e.g. - spermine), the spectral change shown is consistent with this claim albeit in vitro and not cells. Fig. 4 supports the ratiometric claim. Fig. 7 and 8 primarily show a time and dose-dependent change in fluorescence and cell killing. Fig. 9 is weak and of questionable significance. Bottom-line, I agree with reviewer 2 depletion of cellular amine cannot be claimed based on the data shown. The spectroscopic and ratiometric work however are elegantly done. In view of the speculated mechanistic aspect there is sufficient evidence to call this manuscript into question. Minimally I would recommend re-wording the abstract, focusing on the synthesis of a novel photosensitizer nanoparticle formulation with a putative amine depletion anticancer mechanism and to re-write the abstract consistent with the data shown.

Reviewers' comments and Responses

Reviewer #1 (Remarks to the Author):

The changes made by the author are superficial and did not address my concern about polyamine depletion. A 5% decrease in polyamine pools is not considered significant. If one conducts the entire experiment from start to finish three times and averages the results, a standard deviation is encountered as much as 5-10% error, which is what the authors claim as a significant depletion effect. It is not. Perhaps they could conduct the proposed DFMO combination experiment to demonstrate true cellular polyamine depletion and show a synergistic effect with DFMO and their agent. Otherwise, I suggest they drop the claim of polyamine depletion altogether and focus on polyamine detection aspects of their method.

Response: Yes, we agree with the reviewer's comments. We toned down the statement of polyamine depletion and focus on the polyamine detection and photodynamic therapy in the revision.

Reviewer #2 (Remarks to the Author):

To whom it may concern:
I have independently evaluated the manuscript entitled, "Combination of chemical reaction for polyamine detection with photodynamic therapy as well as a depletion of polyamine on cancer treatment and prevention", by Li et al. In the abstract the authors claim, "... new photosensitizer isopyrrolopyrroleaza-BODIPY pyridinium salt 20 (PPAB-PyS) nanoparticles that can react with the over-expressed polyamine in cancer cells and produce two photosensitizers with enhanced phototoxicity on cancer destruction. Meanwhile, PPAB-PyS nanoparticles provide a simultaneous ratio metric fluorescence imaging of intracellular polyamine... This combination ... as well as a depletion of polyamines for cancer treatment and prevention"
Reviewer2 questions the depletion aspect. He/she states, "I am still not satisfied with the data showing these molecules actually reduce polyamine levels in living cells."

In my view, the main figures in a publication must directly support the claims made in the abstract. Two out of three of the reviewers are apparently satisfied, but reviewer2 strongly objects to the third claim "that the use of the nanoparticle photosensitizing agent depletes polyamines". Fig. 1 does support this claim shown with model amine compounds (e.g. –

spermine), the spectral change shown is consistent with this claim albeit in vitro and not cells. Fig. 4 supports the ratiometric claim. Fig. 7 and 8 primarily show a time and dose-dependent change in fluorescence and cell killing. Fig. 9 is weak and of questionable significance. Bottom-line, I agree with reviewer 2 depletion of cellular amine cannot be claimed based on the data shown. The spectroscopic and ratiometric work however are elegantly done. In view of the speculated mechanistic aspect there is sufficient evidence to call this manuscript into question. Minimally I would recommend re-wording the abstract, focusing on the synthesis of a novel photosensitizer nanoparticle formulation with a putative amine depletion anticancer mechanism and to re-write the abstract consistent with the data shown.

Response: Yes, we agree with the reviewer's comments. We did re-wording the abstract, focusing on the synthesis of a novel photosensitizer nanoparticle formulation with a putative amine depletion anticancer mechanism and rewrote the abstract consistent with the data shown.